# Sodium and Potassium Intake, Knowledge Attitudes and Behaviour Towards Salt Consumption Amongst Adults in Podgorica, Montenegro

**DOI:** 10.3390/nu11010160

**Published:** 2019-01-13

**Authors:** Lanfranco D’Elia, Mina Brajović, Aleksandra Klisic, Joao Breda, Jo Jewell, Vuk Cadjenović, Francesco P Cappuccio

**Affiliations:** 1World Health Organization Collaborating Centre for Nutrition, University of Warwick, Coventry CV4 7AL, UK; lanfranco.delia@unina.it; 2Department of Clinical Medicine and Surgery, “Federico II” University of Naples Medical School, 80131 Naples, Italy; 3World Health Organization Regional Office, 81000 Podgorica, Montenegro; brajovicm@who.int; 4Center for Laboratory Diagnostics, Primary Health Care Centre, 81000 Podgorica, Montenegro; aleksandranklisic@gmail.com; 5World Health Organization European Office for Prevention and Control of Noncommunicable Diseases, Moscow 229994, Russia; rodriguesdasilvabred@who.int; 6World Health Organization European Office for Prevention and Control of Noncommunicable Diseases, DK-2100 Copenhagen, Denmark; jewellj@who.int; 7Statistical Office of Montenegro, MONSTAT, 81000 Podgorica, Montenegro; cadjenvicvu@gmail.com; 8Division of Health Sciences, Warwick Medical School, University of Warwick, Coventry CV4 7AL, UK

**Keywords:** Montenegro, salt, sodium, potassium

## Abstract

Excess salt and inadequate potassium intakes are associated with high cardiovascular disease (CVD). In Montenegro, CVD is the leading cause of death and disability. There is no survey that has directly measured salt and potassium consumption in Montenegro. The aim is to estimate population salt and potassium intakes and explore knowledge, attitudes and behaviour (KAB), amongst the adult population of Podgorica. Random samples of adults were obtained from primary care centres. Participants attended a screening including demographic, anthropometric and physical measurements. Dietary salt and potassium intakes were assessed by 24 h urinary sodium (UNa) and potassium (UK) excretions. Creatinine was measured. KAB was collected by questionnaire. Six hundred and thirty-nine (285 men, 25–65 years) were included in the analysis (response rate 63%). Mean UNa was 186.5 (SD 90.3) mmoL/day, equivalent to 11.6 g of salt/day and potassium excretion 62.5 (26.2) mmoL/day, equivalent to 3.2 g/day. Only 7% of them had a salt intake below the World Health Organization (WHO) recommended target of 5 g/day and 13% ate enough potassium (>90 mmoL/day). The majority (86%) knew that high salt causes ill-health. However, only 44% thought it would be useful to reduce consumption. Salt consumption is high and potassium consumption is low, in men and women living in Podgorica.

## 1. Introduction

Non-communicable diseases (NCDs) are the leading, yet preventable, causes of death worldwide [1]. In Montenegro, NCDs are a major public health challenge undermining socio-economic development [2]. According to the World Health Organization (WHO) latest estimates [2], NCDs account for 92% of total deaths, out of which 22% are premature (occurring before the age of 70 years) and 61% due to cardiovascular diseases CVDs. According to the national registry for acute coronary syndrome and cerebrovascular diseases, crude incidence rate of acute coronary syndrome was 176.6 per 100,000 in 2013 and 2007 per 100,000 for cerebrovascular event, respectively [3].

High blood pressure (BP) and unhealthy diet are the leading risk factors for CVD in the world and among the risk factors that account for most of the disease burden in Montenegro [3]. The overall prevalence of raised blood pressure in adults aged 20 and over was 32.7% in 2008 [3]. According to WHO estimates of prevalence of raised blood pressure in 2014 values were higher in men (32.8%) compared to women (23.1%).

A high salt (i.e., sodium chloride, 1 g = 17.1 mmoL of sodium) consumption is an important determinant of high BP. A high salt intake is associated with raised BP, that leads to increased risk of vascular diseases [4]. There is a large body of scientific evidence that high salt intake is associated with raised BP and adverse cardiovascular health [5,6,7,8]. In addition, high salt intake is related to adverse health effects independent of its effects on BP [9,10,11]. Furthermore, a moderate reduction in salt consumption reduces BP [4,5] and it can improve the health outcomes and indirectly reduce the overall mortality through beneficial effect on the BP [7,8].

The World Health Organization (WHO) currently recommends that adults should consume no more than 5 g of salt daily [12]. However, mean daily intakes of salt in most of the countries in the world exceed this recommendation [13,14]. In Montenegro it is common habit to add salt to food at the table and when cooking and to eat processed food high in salt [15]. Since a high proportion of salt derives from hidden sources like processed food, restaurants and other food outlets [16,17], objective assessment of actual salt consumption is necessary and currently lacking in Montenegro. Whilst there is no definitive estimate of population dietary salt intake in Montenegro, it is believed that average consumption could be high, similar to some neighbouring countries in the sub-region, like Serbia (9.85 g/day) [18] and Slovenia [19]; this is far above the dietary target suggested by WHO. Salt reduction strategies in the European region, including Montenegro, encompass monitoring and evaluation actions as one of their important pillars [14,20,21]. Therefore, reliable data on salt intake in Montenegro are needed [22].

A reduction in salt intake is a cost-effective public health action to save lives, to avert diseases and avoid health-care costs [2]. The goal of the WHO is to achieve a 30 per cent reduction in average salt consumption by 2025 [11].

The WHO European Food and Nutrition Action Plan 2013–2020 recommends that countries adopt comprehensive salt reduction strategies [11]. Salt reduction strategies in the European region include monitoring and evaluation as important pillars [14,20]. Hence, comprehensive, reliable data on salt intake in Montenegro are urgently needed, using at least one accurately collected 24 h urine sample for assessing population average salt consumption [21].

In contrast to salt, epidemiological and intervention studies suggest beneficial effects of dietary potassium on BP and cardiovascular health [23,24,25]. As for the salt, Montenegro lacks data also on actual potassium consumption. The WHO currently recommends that adults should consume not less than 90 mmoL of potassium daily [26]. Hence, reliable data on potassium intake in Montenegro are also needed.

The primary aim of the present study was to establish current baseline average consumption of salt and potassium by 24 h urine collection, in a random sample of men and women drawn from the areas surrounding Podgorica, the capital city of Montenegro. The study also aimed to explore knowledge, attitudes and behaviour towards dietary salt for the implementation of a national program of population salt reduction, in line with the WHO global action plan for 2025.

## 2. Materials and Methods

### 2.1. Participants and Recruitment

The sampling frame was the database of registered patients in the Primary Health Centre (PHC) in Podgorica (Montenegro), age 25–65 years. The database was obtained from the Health insurance fund of Montenegro. From the sampling frame and according to PAHO/WHO and EMRO Protocols [27,28] we excluded the following group of patients: people unable to provide informed consent, those with known history of heart or kidney failure, stroke, liver disease, those who recently began therapy with diuretics (less than two weeks), pregnant women, any other conditions that would make 24 h urine collection difficult. The sample for the survey was selected with a sex- and age-stratified random sampling of men and women in six facilities of the PHC of Podgorica (Figure 1).

The survey took place between 15th September and 15th December 2017. From the 1008 households and individuals interviewed in the sampling frame, 639 of them (63%) provided suitable data for inclusion in the survey analysis. Originally, 6 did not meet the inclusion criteria, 99 did not provide date and time of the urine collections, 12 had missing data, 6 provided urine collections with volume less than 500 mL (conventionally taken as not plausible), 237 (23.5%) provided either under-collections (<22 h) or over-collections (>26 h) and 9 had urinary creatinine excretion outside 2 standard deviations (SDs) of the sex-specific distribution of urinary creatinine in the sample (Figure 2).

The excluded participants had similar anthropometric characteristics and blood pressure values but were younger and has faster pulse rate than those included in the final sample (see Appendix A). The survey was carried out in accordance with the Declaration of Helsinki and Good Clinical Practice [29]. Ethical approval for the survey was obtained from the Committee of Research Ethics of the National Centre of Public Health and participants provided written informed consent to take part.

### 2.2. Data Collection

The examination was performed in a quiet and comfortable room, with the participants who were not allowed to smoke, exercise, eat, consume caffeine and to have a full bladder for 30 min before measurements. The survey was carried out in three steps: (a) questionnaire survey, (b) physical measurements and (c) 24 h urine collections.

The questionnaire (face-to-face interview, adapted version of the WHO STEPS Instrument for NCD Risk Factor Surveillance) [30] was used to collect data on respondent’s demographic and socio-economic status (by occupation and educational attainment); diet, frequency of high salt food consumption, fruit and vegetable consumption, knowledge attitudes and behaviour on dietary salt, history of high BP, diabetes and CVDs, lifestyle advice.

Anthropometric indices, BP and heart rate were measured in all participants. Height was measured in cm with a laser stadiometer (SECA 709). Body weight was measured in kg using a digital electronic scale (Transtek, model GBS-721). Body mass index (BMI) was calculated as weight (kg) divided by height squared (m^2^). Overweight was defined as a BMI of 25 kg/m^2^ or more and obesity as a BMI of 30 kg/m^2^ or more. Systolic and diastolic BP and heart rate measurements were taken three times in the right arm on a sitting position, using a universal cuff and automatic BP and heart rate monitors (Boso Medicus Uno, Bosch+Sohn GmbH, Jungingen, Germany). The first measurement was ignored, the mean of second and third measurements being taken for analysis. The measurements were taken after the participant had rested for 15 min and each with three minutes of rest between the measurements (maximum deviation of cuff pressure measurement ± 3 mmHg and of pulse rate display ± 5%). Hypertension is defined as systolic and/or diastolic BP ≥ 140/90 mmHg or regular antihypertensive treatment [31].

A single 24 h urine collection was obtained from the participants. Each participant was given a leaflet with explanations along with the necessary equipment and a record sheet on which participant noted the start and the finish times of their urine collection, any missed urine aliquots and any medication taken during the collection. The participants were carefully instructed on urine collection methodology [27,28]. In an effort to minimize bias, participants were also requested not to change their diet before or during the day of the urine collection. The first void upon waking on the day of collection was discarded. The urine volume of the 24 h collection was measured by field team-members and a urine sample was stored in a cool place for a maximum of 24 h until transportation to the laboratory. Sodium, potassium and creatinine determinations were carried out immediately [27,28]. Sodium and potassium concentration in the urine samples were determined using a Ion Selective Electrode with a Beckman Coulter Synchron CX5PRO System and expressed in mmoL/L [32]. Creatinine concentration was determined through the Creatinine (urinary) Jaffé kinetic method and expressed in mg/dL [33].

### 2.3. Statistical Analysis

All statistical analyses were performed using the SPSS software, version 20 (SPSS Inc., Chicago, IL, USA). To detect approximately 1 g reduction in salt intake over time using 24 h urinary sodium excretion (difference ~20 mmoL/24 h), with a standard deviation of 75 mmoL/day (alpha = 0.05, power = 0.80), a minimum sample of 120 individuals per stratum is recommended [27,28]. Thus, a minimum recommended sample size of 240 was estimated per age and sex groups and adjusted for an anticipated non-response rate of 50% [27,28]. The population was stratified in groups by sex (men and women) and by age (I: 25–34 years, II: 35–49 years, III: 50–65 years). T-test for unpaired samples or analysis of variance (ANOVA) was used to assess differences between group means. Pearson chi-square test was used to test the association between categorical variables. Linear regression analysis was used to detect the association between salt and potassium consumption with socio-economic status, adjusting for age and sex. To convert urinary output into dietary intake, the urinary excretion of sodium (UNa) or potassium (UK) values (mmoL/day) were first converted to mg/day. Then, sodium values were multiplied by 1.05 (assuming that only 95% of sodium ingested is excreted), while potassium values were multiplied by 1.3 (on the assumption that only 70% of the potassium ingested in excreted in the urine [34]. The conversion from dietary sodium (Na) intake to salt (NaCl) intake was made by multiplying the sodium value by 2.542. The results were reported, as appropriate, as mean (SD and/or 95%CI) or as percentages. Two-sided p below 0.05 were considered statistically significant.

## 3. Results

The population sample included 639 participants between 25 and 65 years old (*n* = 285 or 45% men and *n* = 354 or 55% women), recruited from six facilities (29.3% from Centar, 14.6% from Nova Varos, 9.2% from Stara Varos, 23.3% from Block 5, 11.9% from Zagoric and 11.7% from Tolosi) (Figure 1).

### 3.1. Characteristics of the Participants

The characteristics of the participants are shown in Table 1. There was no statistically significant difference in the mean age between male and female participants, however men were significantly taller and heavier than women and had a higher systolic, diastolic BP and pulse rate. The prevalence of hypertension was on average 36.4% (225/618), comparable in men (110/279 or 39.4%) and women (115/339 or 33.9%; *P* > 0.05).

### 3.2. Daily Urinary Excretions of Volume, Sodium, Potassium and Creatinine and Salt and Potassium Intake

Average urinary volume excretion was 1629 mL per day, being higher in men than women (Table 2). Average urinary creatinine excretion was 1.41 g per day, being again higher in men than women (Table 2). Urinary sodium excretion showed a normal distribution with a tail skewed to the right (i.e., towards higher values). Mean urinary sodium was 186.5 (SD 90.3) mmoL/24 h (Table 2), equivalent to a mean consumption of 11.6 (5.6) g of salt per day (Table 2). Men excreted more sodium than women (mean difference 63.9 mmoL/24 h, *P* < 0.001), equivalent to 4.0 g of higher salt consumption than women. Only 46 participants (7%) met the levels of salt intake of 5g or less recommended by the WHO, significantly less in men than women (*n* = 11 or 4% vs *n* = 35 or 10%, respectively, *P* = 0.003). Urinary potassium excretion showed a normal distribution with a tail skewed to the right (i.e. towards higher values). Mean urinary sodium was 62.5 (26.2) (Table 2), equivalent to a mean consumption of 3.2 (1.3) g of potassium per day (Table 2).

Men excreted more potassium than women (mean difference 10.9 mmoL/24 h, *P* < 0.001), equivalent to 0.6 g of higher potassium consumption than women. Only 79 participants (13%) met the levels of potassium intake of 90 mmoL/day or more recommended by the WHO, significantly less in women than men (*n* = 26 or 8% vs *n* = 53 or 19%, respectively, *P* < 0.0001). No differences in salt and potassium consumption were detected by age (see Appendix A). Salt and potassium consumption varied by socio-economic status, both measured by educational attainment and by occupation (Figure 3). After adjustment of age and sex, salt consumption was significantly lower in participants with the highest educational attainment (*P* for trend = 0.02) and it was the highest in retired and unemployed (*P* for trend = 0.02), suggesting a relationship between high salt intake and low socio-economic status. On the contrary, potassium consumption showed a direct relationship with educational attainment (*P* for trend = 0.05) and with current employment (*P* for trend = 0.05), suggesting a lower potassium consumption in more disadvantaged socio-economic groups.

### 3.3. Knowledge, Attitude and Behaviours Towards Salt Intake

A high proportion of participants (73.8%) reported adding salt to food at the table often or always, more amongst women compared to men (79.1 vs 67.2%; *P* = 0.0007) (Table 3). However, only 21.4% said they would often or always add it when cooking, especially men (25.2 vs 18.3%; *P* = 0.03). The majority (85.8%) knew that high salt causes serious health problems. However, only 44.2% thought it would be useful to reduce salt consumption, 35.6% felt their intake was excessive and 34.3% were doing anything to try and reduce it. Women were more pro-active than men in doing so (39.8 vs 27.6%; *P* = 0.001).

## 4. Discussion

This is the first survey on salt and potassium consumption carried out in adults in Montenegro, using the gold standard measure of 24 h urinary sodium and potassium excretions as biomarkers of intake. The results show unequivocally that salt consumption is high and that potassium consumption is low, both in men and in women.

Average salt intake exceeded by over twofold the WHO recommended maximum population target of 5 g per day [12]. Only 7% of the participants met the WHO salt targets. Men excreted more sodium than women. Discretionary use of salt is very common in Montenegro, with 3 out of 4 participants adding salt regularly to food and 1 in 4 also using it regularly when cooking. The majority of participants knew that high salt causes serious health problems. However, only less than half thought it would be useful to reduce its consumption, 1 in 3 felt own intake was not excessive and only 1 in 3 was reporting doing anything to reduce it. The answers to these questions reveal an insufficient level of knowledge of the real problem associated with high salt consumption amongst the participants of Podgorica and the widespread unreadiness to transfer this knowledge to behavioural changes in using discretionary salt. Average potassium intake was low in Montenegro, less than half the WHO recommended minimum population target of 90 mmoL/day or more [23]. Only 13% met WHO potassium targets.

Our main findings point out that the salt intake in Montenegro is higher than those reported in many neighbouring countries, both in men and women [35,36]. In the recent MINISAL study, the daily urinary sodium excretion of Italians was 189 for men and 147 mmoL/24 h for women [34] and in Greece 194 and 158 mmoL/24 h for male and female participants, respectively [36]. On the other hand, the potassium intake was similar to that found in Greece [36] but higher than that measured in Italy [35]. As reported in the majority of previous studies [37], men consume more salt than women. This is almost certainly explained by greater total food consumption. As expected, there are gender differences in both salt and potassium consumption, as detected in many other population surveys. Salt and potassium are expressed as total quantities rather that consumption per calorie intake, hence the gender difference is mainly explained by the larger body size of men compared to women and the corresponding total food consumption compared to women. However, there were gender differences in attitudes and behaviours towards salt consumption. Women reported more often less use of discretionary salt and more readiness to doing something to control salt intake. These could contribute, at least in part, to the differences found.

Social inequalities are important determinants of ill health globally [38,39]. People from low socio-economic background die sooner and have more disabilities. They are more likely to depend of cheaper, unhealthy food rich in salt. In the UK, their knowledge of public health recommendations is lower and the use of table salt and total salt intake is higher [34,40,41]. In England and Wales, the salt reduction of 1.4 g per day seen recently has not resulted in a reduction in socioeconomic differences [42]. “Upstream” health policies tend to reduce health inequalities [43]. Furthermore, an improvement in socio-economic position decreases the risk hypertension that is seen with low parental social status [44].

Socioeconomic status is a major determinant of excessive dietary salt and insufficient potassium intake [34,41,42]. Indeed, the healthy foods are more expensive than Western dietary patterns [45], that is, more energy-dense and nutrient-poor foods [46], that leads to a high availability of poor-quality foods. Also, low level of education unfavourably contributes to higher salt and lower potassium intake [34,41,42,47]. Hence, the interaction of low income and low levels of education contribute together to a low adherence to healthy eating pattern. Our results confirm this trend even in Montenegro, where unemployed subjects or participants with low level of education consumed more salt and less potassium. These conditions, in turn, would contribute to increase the otherwise avoidable burden of CVD. Currently, in the European region, a gradient across measures of inequalities is present even for the degree of implementation of current salt reduction policies [48].

### 4.1. Strengths and Limitations

Strengths of our study are the inclusion of a large random sample of male and female participants as representative of the capital city of Montenegro, the assessment of average consumption of salt and potassium by 24 h urine collection (gold-standard measure of salt and potassium intake) [28], the rigorous quality control and the careful standardized protocol of urine collections. In particular, the rigorous instructions to ensure completeness of urine collections and the strict protocol to select for analysis only those fulfilling the quality control criteria among which the length of collection time and the assessment of urinary creatinine excretion, markers of the accuracy of the collection. However, the study has some limitations. The possibility of selection bias cannot be excluded. Participants not delivering complete urines were younger than those complying. However, no other differences were seen in body size and cardiovascular parameters. Urinary sodium and potassium excretions were only assessed once. Given the large within-subject variability in consumption, it is acknowledged that a single collection is unlikely to characterize an individual’s salt and potassium consumption [49]. However, group estimates are less likely to be biased by this variability. Finally, sampling was drawn from the capital city of Podgorica. Whilst this represents a valid random sample, it may not be fully representative of the entire country, consisting of a mountainous Northern Region and a small Coastal Region on the Adriatic Sea. Montenegro is a small country with an estimated population of approximately 640,000, of which approximately 150,000 resides in the capital and surroundings [50]. The recent and continuous migration to the capital city makes the population of Podgorica rather heterogeneous, although some differences in dietary habits between the different regions may still exist.

### 4.2. Conclusions

Salt intake is high and potassium is low in Podgorica, Montenegro. Awareness, attitudes and behaviours about salt and its implication for health suggest that there is an urgent need for intensive awareness campaigns and health promotion to improve the take up of preventive strategies aiming at reducing salt consumption, whilst at the same time increasing potassium intake by encouraging higher consumption of potassium-rich food. Awareness about hidden salt in processed food should be highlighted. A national program for reducing salt intake in Montenegro needs to be implemented through systematic efforts and multisectoral collaboration including not only public awareness and behaviour-change communication (including via health care professionals), but, more importantly, structured programs for reformulation that set the framework for the food industry to reduce salt in processed foods, major source of salt intake. The current Montenegrin approach to reducing population salt intake is based predominantly upon voluntary food supply reformulation and the Government has already introduced a regulation on maximum salt content in bakery products. Based on international experience and using national dietary intake data, the government can set food category-based salt reduction targets for the most important foods that contribute to salt intake among the Montenegrin population, with the objective of contributing to a 30% reduction in salt consumption by 2025. Only one in two people consume sufficient amount of potassium and public policies should be directed to encouraging an increase intake of fruit, vegetable, pulses and nuts by setting targets to increase consumption to 90 mmoL per day or more. In the absence of a more comprehensive national survey of habitual salt and potassium intake in Montenegro, our data provides a useful baseline against which to monitor the impact of future initiatives.

## Figures and Tables

**Figure 1 nutrients-11-00160-f001:**
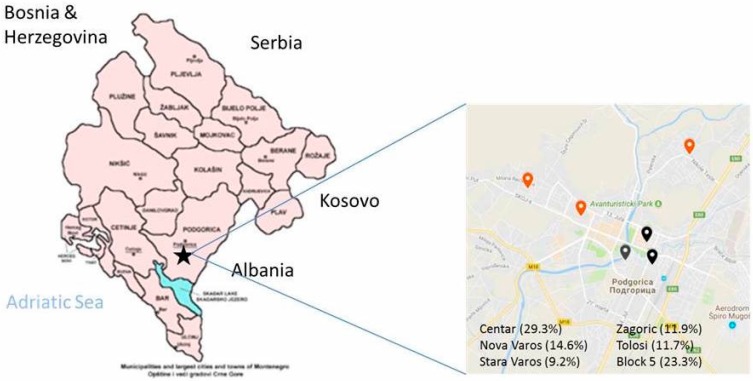
Geographical sampling from Podgorica.

**Figure 2 nutrients-11-00160-f002:**
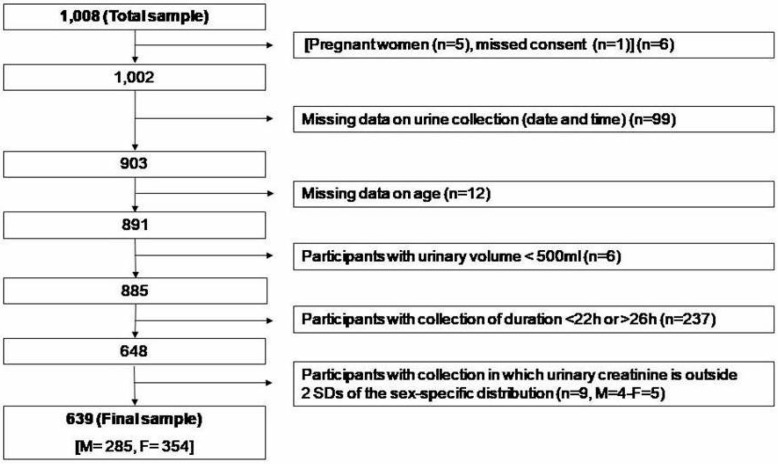
Stepwise procedure for the selection of valid participants according to protocol adherence, quality control and completeness of 24 h urine collections.

**Figure 3 nutrients-11-00160-f003:**
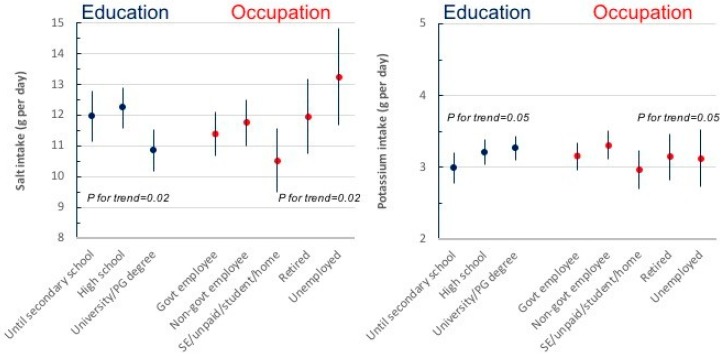
Association between salt and potassium consumption and socio-economic status, defined by educational attainment or occupation, after adjustment for age and sex (results are mean ± SD; *P* values for linear trend are reported).

**Table 1 nutrients-11-00160-t001:** Characteristics of the participants.

	All (*n* = 639)	Men (*n* = 285)	Women (*n* = 354)
Age (yrs)	46.8 (11.3)	46.7 (11.3)	46.9 (11.3)
Height (cm)	175.4 (9.2)	182.5 (7.0)	169.6 (6.1) ^†^
Weight (kg)	79.8 (16.8)	90.9 (15.2)	70.7 (12.0) ^†^
BMI (kg/m^2^)	25.8 (4.2)	27.1 (3.8)	24.6 (4.3) ^†^
Systolic blood pressure (mmHg)	125.1 (16.2)	129.2 (14.4)	121.8 (16.7) ^†^
Diastolic blood pressure (mmHg)	79.1 (9.2)	80.7 (7.6)	77.9 (10.1) ^†^
Pulse Rate (b/min)	72.4 (8.7)	73.3 (8.5)	71.6 (8.9) ^†^
Current smokers *n* (%)	166 (26.2)	68 (23.9)	98 (28.1)
Occupation* (in the last 12 months) *n* (%)			
Government employee	195 (30.6)	74 (26.0)	121 (34.5) ^†^
Non-government employee	188 (29.5)	99 (34.7)	89 (25.4)
Self-employed	47 (7.4)	32 (11.2)	15 (4.3)
Unpaid	6 (0.9)	4 (1.4)	2 (0.6)
Student	3 (0.5)	2 (0.7)	1 (0.3)
Housewife	39 (6.1)	-	39 (11.1)
Retired	80 (12.6)	35 (12.3)	45 (12.8)
Unemployed (able to work)	36 (5.7)	18 (6.3)	18 (5.1)
Unemployed (unable to work)	5 (0.8)	3 (1.1)	2 (0.6)
Other	11 (1.7)	6 (2.1)	5 (1.4)
Refused	26 (4.1)	12 (4.2)	14 (4.0)
Education attainment * *n* (%)			
Less than primary school	6 (0.9)	2 (0.7)	4 (1.1)
Primary school	24 (3.8)	9 (3.2)	15 (4.3)
Secondary school	128 (20.1)	57 (20.0)	71 (20.2)
High school	237 (37.3)	119 (41.8)	118 (33.6)
College/University	201 (31.6)	81 (28.4)	120 (34.2)
Postgraduate degree	26 (4.1)	12 (4.2)	14 (4.0)
Refused	14 (2.2)	5 (1.8)	9 (2.6)

Results are mean (SD) or as percentage; ^†^
*P* < 0.001 vs men; * *n* = 636.

**Table 2 nutrients-11-00160-t002:** Daily urinary excretions of volume, sodium, potassium and creatinine and estimates of salt and potassium intake.

	All (*n* = 639)	Men (*n* = 285)	Women (*n* = 354)
Volume (mL/24 h)	1629 (608)	1694 (569)	1576 (633) *
Sodium (mmoL/24 h)	186.5 (90.3)	221.9 (101.3)	158.0 (68.2) ^†^
Potassium (mmoL/24 h)	62.5 (26.2)	68.5 (27.6)	57.6 (23.9) ^†^
Creatinine (g/24 h)	1.41 (0.55)	1.73 (0.60)	1.15 (0.35) ^†^
Salt intake (g/day)	11.6 (5.6)	13.9 (6.3)	9.9 (4.3) ^†^
Potassium intake (g/day)	3.2 (1.3)	3.5 (1.4)	2.9 (1.2) ^†^

Results are mean (SD); ^†^
*P* < 0.001; * *P* < 0.02 vs men.

**Table 3 nutrients-11-00160-t003:** Knowledge, attitudes and behaviours towards salt consumption.

Question	Total (*n* = 639)	Men (*n* = 285)	Women (*n* = 354)
Do you add salt to food at the table? (Often/Always)	73.8%	79.1%	67.2% *
In the food you eat at home salt is added in cooking (Often/Always)	21.4%	25.2%	18.3% **
How much salt do you think you consume? (Far too much/Too much)	35.6%	37.5%	34.0%
Do you think that a high salt diet could cause a serious health problem? (Yes)	85.8%	82.8%	88.3%
How important is lowering salt/sodium in your diet? (Very important)	44.2%	40.5%	47.1%
Do you do anything on a regular basis to control your salt/sodium intake? (Yes)	34.3%	27.6%	39.8% *

Results are percentages * *P* < 0.001; ** *P* = 0.03 vs men.

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
