# Peer review of "Sodium and Potassium Intake, Knowledge Attitudes and Behaviour Towards Salt Consumption Amongst Adults in Podgorica, Montenegro"

_nutrients, 2019, doi:10.3390/nu11010160_

Reviewer 1 Report

This manuscript deals with the sodium and potassium salt consumption in Montenegro. It explores also knowledge, attitude and behaviours of the population toward salt intake and discusses accordingly. This manuscript is well written and bring relevant information about sodium and potassium consumption.

Just few remarks:

- Few information are given on the customary dietary consumption (type of foods, composition, …).

- L182-183 : It could be shortly explained why this multiplication.

- According to the presented data, it seems that population in Montenegro is a strong consumer of sodium with excessive sodium intake in the diet. I am not a specialist neither in nutrition nor in socio-demographic studies but it seems to me that this could be the opportunity to make comparison on different and various criteria (consumption behaviour, nutrition, physiology…) with comparable data obtained from countries which are lower sodium consumer or near the WHO recommendations.

- How the difference is sodium urinary excretion between sexes can be explained. Can this be explained by metabolic or physiological differences or differences in consumption behaviour?

Author Response

We thank the reviewer.

Detailed dietary assessment was beyond the scope of this study, which was focusing on salt. KAB questionnaire gave sufficient indications to set priorities for policy actions as suggested by the WHO protocols (Ref. 23-24)

L182-183: added clarifications to the text

Montenegrin intake is not much different from countries nearby. No country in the world are at or near WHO recommendations.

We devote an extended paragraph (L262-269) to explain the reasons for the differences in salt intake between men and women, that are not surprising. We express intake in absolute terms and the consumption of salt depends on the consumption of food by weight, and men eat more food than women.

Reviewer 2 Report

Dear Editor, 

I am pleased to review the assigned manuscript. Overall, a proposed manuscript looks good to me. Authors really did a wonderful job and presented very nice & relevant literature. The English is generally satisfactory, although there are some places where corrections and/or changes are required. The study does have few minor issues. The authors should improve the paper following these suggestions.

·         Abstract look good to me, it is covering the theme of whole manuscript

·         The introduction should be better organized. Some of the sentences are not well structured, should be clarified and rewritten. It is advice to link the story in a better way in an introduction to convey a proper message to readers.

·         Please add few suggested and latest references in background section

-       A Question Mark on Iron Deficiency in 185 Million People of Pakistan: Its Outcomes and Prevention.

-       Concept of double salt fortification; a tool to curtail micronutrient deficiencies and improve human health status.

·         Result and discussion section is good-  it’s well written and well organized

·         I would suggest authors to improve figures especially figure 2 and 3

·         Conclusion looks good to me, please remove the reference from conclusion

·         Please recheck the reference style, some of the references are not according to the journal instructions.

Author Response

We thank the reviewer.

Abstract unchanged.

We have reviewed the language and improved where we felt there was the need.

The Introduction has been extensively amended and reduced: the structure follows the logic - NCD and CVD epi; high BP as cause of CVD; high salt as cause of high BP hence high CVD; population approach and policy actions; role of potassium; aims.

We are not sure we understand the second point of adding references on Iron deficiency in Pakistan and of double fortification of salt. We have not measured iodine (or iron) in this study and it is difficult to start speculating beyond the scope of the study.

The discussion has been modified in response to other reviewers and the editor.

We are unable to improve the quality of Fig 2 and 3 - we shall seek advice at the productions stage

Reference 47 from conclusions removed as suggested

We have corrected the few inconsistencies with requested reference style.

Reviewer 3 Report

I have revised the manuscript entitled “Sodium and potassium intake, knowledge attitudes and behaviour towards salt consumption, amongst adults in Podgorica, Montenegro”. The study has two aims: First, to establish current baseline average consumption of salt and potassium by 24h urine collection, and second, to explore knowledge, attitudes and behaviour towards dietary salt in Montenegro. In my opinion, this study is a work of great interest, given the lack of data on sodium intake in adults in Montenegro, and the implications of the high consumption of salt on health. On the other hand, there are not many studies that have addressed the study of the attitude of the population on salt consumption, and this is a very necessary aspect to be able to design and implement successful programs to reduce salt consumption. The results are presented appropriately and are very relevant, and they will serve as a starting point to monitor the consumption of salt in this population. The manuscript is well written and organized, although there are some minor points that can be improved.

L5-7: In my opinion the numbers that correspond to the authors' affiliation should be superscripts

Introduction:

L68-69: “In Montenegro it is common habit to add salt to food at 68 the table and when cooking and to eat processed food high in salt” this sentence needs a reference

L69-70: “Since a high proportion of salt derives from hidden sources like processed food, restaurants and other food outlets…” needs a reference

L72-73:”… it is believed that average consumption could be as high as 14g per day, similar to the neighboring countries in the sub-region.” needs a reference. Explain also which countries you are referring to.

L101-102: “gold-standard measure of salt and potassium intake” needs a reference (both for sodium and potassium)

Materials and methods

-Explain the database used to perform the sampling in more detail. How old were the patients registered in the database? Has any age range been considered as an inclusion / exclusion criterion in this study? Although reference is made to "inclusion criteria" (L19) they are not clearly indicated in the text.

L121: It appears that providing less than 500 mL of urine is an exclusion criterion, but it would be necessary to justify this cut off point or clarify if it has been established arbitrarily.

L145-146: ”Body weight was measured in kg and height in cm with electronic scales” provide the model of the instruments used

Results

Table 1.- Review the Occupation and Education data so that the categories are on the same line as the data

L207: Potassium has not been determined in all subjects, explain the reason.

Discussion

L328: Dairy products are also a good source of potassium, and milk and yoghurts provide little sodium. However, the authors do not take into account this food group among the recommendations to be taken into account in public policies. Is dairy products intake in Montenegro is high enough to justify this omission?

 Author Response

We thank the reviewer.

The aims are clearly spelt out in the last paragraph of the Introduction.

Table 1. Alignments corrected

Table 2. We refrained deliberately from analyzing the data on an individual basis, using for instance correlations or regressions. The study design (and subsequent sample size calculations) were made on the assumption of establishing 'group' means with precision rather that assessing participants' individual habitual sodium and potassium intake. A single 24h urine collection, whilst useful in assessing population group averages (for monitoring effectiveness of population programmes), is inadequate to characterize an individual habitual salt intake.

Likewise for relationships with BP

Reviewer 4 Report

In this manuscript (ID nutrients-408329), titled “sodium and potassium intake, knowledge attitudes and behavior towards salt consumption, amongst adults in Podgorica, Montenegro”, authors estimate salt and potassium intakes and explored knowkedge, attitudes, and behavious amongst the adult population of Podgorica. The results of this study indicate that salt consumption is high, and potassium consumption is low in the population of Podgorica as compared with WHO recommendations. My major concern is that the research aim is missing. I could not know the scientific problem that this study is trying to answer. This manuscript looks like a government report instead of scientific investigation. Several minor concerns are listed in the following. 

1. In table 1, the number in each row in the Occupation section should be lined up with its corresponding title. The same problem is as to the Education attainment section. 

2. In table 2, the urinary volume, sodium, potassium of urinary excretions were compared between men and women. These parameters are all related with body weight (or body size). I think that these parameters should be normalized to their body weight in order to be compared between men and women to rule out the effect of body weight (or body size). 

3. Is the salt or potassium intake related with their blood pressure in this population?

 Author Response

We thank the reviewer.

L5-7 - done

L68-69: New ref 16 added

L69-70: New ref 17 added

L72-73: we have now made it clear referring to Serbia and Slovenia with additional references 18 and 19 (

L101-102: sentence removed

Clarification has been added under sampling that there was an age restriction to 25-65 years for selection of registered patients.

L121: a urinary volume of less than 500mL in temperate climates is conventionally considered implausible and in this particular case is only excluding additional 6 participants. Our approach to quality control of 24h urine collection has been highly conservative.

L145-146: details provided for height, weight and BP measurements devices

Table 1: aligned now

L207: amended details in table due to transcription error

L328: We feel that to extend recommendations as suggested without more direct nutritional assessments would be beyond the scope of our study. We are not sure whether the small amount of potassium present in milk and yoghurts would be a major contributor to i9ncreasing potassium consumption, unless large quantities were recommended.

 Round  2

Reviewer 4 Report

no additional concern